# Deep Sensing: Active Sensing using Multi-directional Recurrent Neural Networks

**Jinsung Yoon**
Department of Electrical and Computer Engineering
University of California, Los Angeles
Los Angeles, CA 90095, USA
jsyoon0823@g.ucla.edu

**William R. Zame**
Department of Mathematics
Department of Economics
University of California, Los Angeles
Los Angeles, CA 90095, USA
zame@econ.ucla.edu

**Mihaela van der Schaar**
Department of Engineering Science, University of Oxford, Oxford, UK
Alan Turing Institute, London, UK
mihaela.vanderschaar@eng.ox.ac.uk

## Abstract

For every prediction we might wish to make, we must decide *what* to observe (what source of information) and *when* to observe it. Because making observations is costly, this decision must trade off the value of information against the cost of observation. Making observations (sensing) should be an active choice. To solve the problem of active sensing we develop a novel deep learning architecture: *Deep Sensing*. At training time, Deep Sensing learns how to issue predictions at various cost-performance points. To do this, it creates a different presentation at each of a variety of different performance levels, each associated with a particular set of measurement rates (costs). This requires learning how to estimate the value of real measurements vs. inferred measurements, which in turn requires learning how to infer missing (unobserved) measurements. To infer missing measurements, we develop a Multi-directional Recurrent Neural Network (M-RNN). An M-RNN differs from a bi-directional RNN in that it sequentially operates *across streams* in addition to *within streams*, and because the timing of inputs into the hidden layers is both lagged and advanced. At runtime, the operator prescribes a performance level or a cost constraint, and Deep Sensing determines what measurements to take and what to infer from those measurements, and then issues predictions. To demonstrate the power of our method, we apply it to two real-world medical datasets with significantly improved performance.

## 1 Introduction

Making observations is costly. Hence, for every prediction we might wish to make, we must decide what to observe – i.e., what source of information to consult/use – and when to observe it. This (joint) decision involves a trade-off between the value of the information that will/might be obtained from the observation and the cost of making the observation. There is little reason to make an observation if the result of that observation can already be confidently estimated on the basis of what is already known or if the result would be of little value in any case; it would be much better to conserve the resources to make a different observation at a different time. Thus *making observations (sensing) should be an active choice* (Yu et al. (2009)). The problem of active sensing has many applications, from healthcare (the example we use here to illustrate our method) to neuroscience to robotics to wireless communications.

The central point of our approach is that we need to estimate the value of information. This must be learned at training time. We learn the estimated value for a specified set of measurements by first predicting on the basis of the information we have, then deleting the specified set of measurements, inferring what we have deleted on the basis of the data that remains, making a new prediction on

the basis of the inferred measurements and the remaining data, and comparing the two predictions. (Part of our architecture is designed specifically for these tasks.)

To infer missing data, we develop a novel architecture called a Multi-directional Recurrent Neural Network (M-RNN). Like a bi-directional RNN (Bi-RNN) (Graves & Schmidhuber (2005)), an M-RNN operates forward and backward in each data stream - in the *intra-stream directions*. Unlike a Bi-RNN, an M-RNN also operates *across* streams – in the *inter-stream directions*. And unlike a Bi-RNN, the timing of inputs into the hidden layers of an M-RNN is lagged in the forward direction and advanced in the backward direction. (To the best of our knowledge, our architecture is the first that operates in this way). Our M-RNN executes both interpolation (intra-stream) and imputation (inter-stream) to infer missing data.

Because we need to trade off performance against cost, our neural network must learn – at training time – how to issue predictions at various cost-performance points. To do this, it creates multiple presentations (neural network parameters) at various performance levels associated with different measurement rates (costs). Each presentation is learned on the basis of a particular set of missing data; these sets are constructed recursively and adaptively.

An important aspect of our solution to the active sensing problem is that there are differences between the operation at training time and runtime. At training time we can use data at the current time to infer missing data at an earlier time; i.e. we can operate *non-causally*. We cannot do so – and do not do so – at runtime. However, after a new sample *is* received at runtime, we can and do go back to improve the previous inferences (interpolations and imputations) which will in turn improve both current and future predictions.

To demonstrate the power of our method, we apply it to two real-world medical datasets. We show that our method yields significantly greater predictive power (measured as the Area Under the ROC Curve (AUC)) per unit cost in comparison to other state-of-the-art methods. Because our inference methods are of interest in themselves, we compare the root mean squared error (RMSE) and corresponding AUC for our method to that of state-of-the-art imputation methods in statistics such as White et al. (2011); Rehfeld et al. (2011); García-Laencina et al. (2010), RNN-based imputation methods such as Choi et al. (2015); Lipton et al. (2016); Che et al. (2016); Futoma et al. (2017), and interpolation methods such as Kreindler & Lumsden (2012); Mondal & Percival (2010). In all cases, we demonstrate large and significant improvements.

## 2    RELATED WORKS

Previous works related to Deep Sensing fall into four areas: Active sensing, missing value inference, Bayesian optimization and RNN methods. (Active sensing is related to both active learning and to reinforcement learning, but actually rather different from both of them; see the discussion below.)

**Active sensing** As discussed in the Introduction, the focus of active sensing (Yu et al. (2009); Alaa & van der Schaar (2016)) and of screening policies (Ahuja et al. (2017)) is to determine what and when to measure; this is an important question whenever acquiring measurements is costly. Yu et al. (2009) studies the problem of active sensing using a Bayesian approach with Gaussian processes. This work models the data streams as Gaussian processes, so if the number of data streams is $D$ then the number of parameters is of order $D^2$ and estimation accuracy decreases dramatically as $D$ grows. Moreover, this work creates only a single presentation for the entire data set and hence does not effectively trade off predictive gain against measurement cost, and cannot deal with a setting in which there are different costs to sample different variables. Alaa & van der Schaar (2016) addresses the problem of active sensing for a single data stream. That work assumes a specific stochastic process to learn the optimal time to sample the next measurement, given the characteristics of the specific stochastic process. Because this work treats only a single data stream and imposes a particular model of the data, this approach cannot be applied to a general data stream and is ineffective in active sensing across multiple data streams. Ahuja et al. (2017) proposes a methodology for personalized screening in the medical domain but the procedure for learning presentations is independent of the screening policy.

A particular approach to active sensing (*submodular optimization*; see Iwata et al. (2001); Schrijver (2000) for example) minimizes a submodular objective function as a proxy for minimizing the true "cost - information gain". Deep Sensing does not use a submodular objective function – or any

other particular objective function – as a proxy; instead, Deep Sensing uses a greedy algorithm to find the individual measurements that yield a positive "information gain - cost", and uses the set of all such individual measurements as the set of measurements that should be performed. (If this set is empty, Deep Sensing moves on to the next possible measurement date, and so forth.) The details are discussed in Section 3.

**Missing value inference:** There are two standard methods to deal with missing information in time-series data streams: interpolation and imputation. Interpolation methods (Kreindler & Lumsden (2012); Mondal & Percival (2010)) attempt to capture the temporal relationships within each data stream but not the relationships across streams. Imputation methods (White et al. (2011); Rehfeld et al. (2011); García-Laencina et al. (2010)) attempt to capture the synchronous relationships across data streams but not the temporal relationships within streams. (Most of this work assumes a specific model of the data, rather than learning a presentation from the data, as Deep Sensing does.) We are not aware of any previous work that attempts to capture both the relationships *within stream* and the relationships *across streams*.

**Bayesian optimization:** The problem of costly measurements has been studied in other areas as well. Bayesian optimization (Pelikan et al. (1999); Snoek et al. (2012)) uses a Gaussian process regression (GPR) to approximate the loss function for a given optimization problem (Seo et al. (2000)). This approximation is then used to sequentially evaluate the true loss function at points where the expected decrease in loss is the greatest. When function evaluations are computationally costly (e.g. for hyper-parameter optimization in complex problems), this approach is a way of identifying good minima given constraints on time.

There are significant differences between Bayesian optimization and Deep Sensing. Firstly, in Bayesian optimization, "cost" is usually taken to be computation time, and optimization is performed subject to a cost constraint – a maximum number of permissible evaluations. In traditional settings, this cost is essentially treated as constant, and not explicitly considered in selecting points during the optimization procedure. Deep Sensing, on the other hand, *trades off* cost against (predictive) gain. Secondly, in the active sensing setting we consider, measurements can be taken only forwards in time; in the setting of Bayesian optimization, no restrictions are placed on the location of function evaluations. Active sensing thus captures the problems in the healthcare setting, in which causal predictions are needed to inform the actions of practitioners in a timely fashion. Finally, Bayesian optimization uses GPR to approximate loss functions, which places limitations on the types of functions which it can mimic. Because neural networks are "universal approximators" (Hornik et al. (1989)), the RNNs used in Deep Sensing allow it to model a richer set of functions (and give rise to more complicated and interesting dynamics).

**RNN methods:** RNNs have been used successfully for prediction on the basis of time-series data with missing data and irregular sampling. The approach of Gingras & Bengio (1996) is to first replace all the missing information with a mean value and use the feedback loop from the hidden states to update the imputed value while learning the classification problem using a standard RNN. Tresp & Briegel (1998) uses the Expectation-Maximization (EM) algorithm to impute the missing values and uses the reconstructed data streams as inputs to a standard RNN for prediction. As with standard imputation methods, the imputation depends only on the synchronous relationships across data streams and not on the temporal relationships within streams. Parveen & Green (2002) use a linear model to estimate missing values from the latest measurement and the hidden state of each stream. As with standard interpolation methods, the estimate depends only on the temporal relationships within each stream and not on the relationships across streams.

More recent works address both missing values and irregularly sampled time-series data streams (Choi et al. (2015); Lipton et al. (2016); Che et al. (2016); Kim et al. (2017)). These papers use the sampling times to capture the informative missingness and time interval information to deal with irregular sampling. They do this by concatenating the measurements, sampling information and time intervals and using the concatenation as the input of an RNN. These papers differ in the replacements they use for missing values. Choi et al. (2015); Lipton et al. (2016); Kim et al. (2017) replace the missing values with 0, mean values or latest measurements – all of which are independent of either the intra-stream or inter-stream relationships or both. Therefore, those methods cannot be extended to our active sensing algorithm. Che et al. (2016) imputes the missing values using only the latest measurements, the mean value of each stream, and the time interval. It is not bi-directional and so cannot use information available at a given time to update estimates of information that is missing at

an earlier time. Futoma et al. (2017) assume a Gaussian Process in order to learn the latent variables from irregularly sampled longitudinal datasets, and use the outputs of this Gaussian Process as the inputs of an RNN to deal with the irregular sampling of the dataset.

**Active learning and reinforcement learning** Active learning (e.g. MacKay (1992); Seung et al. (1992)) and reinforcement learning both have something in common with active sensing, in that they all have to do with the (possibly costly) acquisition of information. However active learning focuses on the acquisition of *labels*, while active sensing focuses on the (costly) acquisition of *measurements*. And reinforcement learning focuses on *actions* (which directly affect the state), while active sensing focuses on *observations* (which do not affect the state).

# 3 BACKGROUND

## 3.1 NOTATION

The training set consists of $N$ arrays of data. It is convenient to use the language of healthcare and to speak of the array $n$ as the information of patient $n$, so that there are $N$ patients in the training set. For each patient $n$, we have a multivariate time-series data stream of length $T$ (the length $T$ and the other components may depend on the patient $n$ but for the moment we suppress the dependence on $n$) that consists of three components: measurements $\mathcal{X}$, labels $\mathcal{Y}$ and time stamps $\mathcal{S}$.

Because measurements are not necessarily made at regular intervals, we distinguish between time stamps and actual times. The *time stamp* $t = 1, 2, \ldots$ simply indexes the sequence of times at which measurements were taken; $s_t$ is the actual time at which the measurements $x_t$ were taken and the label $y_t$ was realized. For convenience we normalize so that $s_1 = 0$; we assume actual times are strictly increasing: $s_{t+1} > s_t$ for $0 < t \leq T - 1$.

The *label* $y_t$ represents the outcome realized for patient $n$ at time stamp $t$ (actual time $s_t$). Labels may be discrete or continuous. In the former case we are considering a classification problem (e.g. prediction of an event, such as discharge, clinical deterioration, death); in the latter case, we are considering a regression problem (e.g. prediction of value or family of values). If we are interested explicitly in the estimation of missing data for its own sake, then $y_t$ would represent the actual observed data at time stamp $t$. $\mathcal{Y}$ is the vector of outcomes for this patient. We normalize so that labels and predictions lie in $[0, 1]$.

There are $D$ streams of measurements; each measurement is a real number, but not all measurements may be observed at each time stamp. Hence we view the set of possible measurements at time stamp $t$ as $\mathbb{R}_* = \mathbb{R} \cup \{*\}$. We interpret $x_t^d = *$ to mean that the stream $d$ was not measured at time stamp $t$; otherwise $x_t^d \in \mathbb{R}$ is the measurement of stream $d$ at time stamp $t$. $\mathcal{X}$ is the array of measurements of all streams at all time stamps for the patient under consideration.

It is convenient to introduce some notation to keep track of what is measured/not measured (observed/not observed). For each time stamp $t$ and stream $d$, write $m_t^d = 0$ if $x_t^d = *$ (not measured) and $m_t^d = 1$ if $x_t^d \in \mathbb{R}$ (measured). Let $\delta_t^d$ be the actual amount of time that has elapsed since the stream $d$ was measured last. $\delta_t^d$ can be defined recursively as follows:

$$\delta_t^d = \begin{cases} s_t - s_{t-1} + \delta_{t-1}^d & \text{if } t > 1, m_{t-1}^d = 0. \\ s_t - s_{t-1} & \text{if } t > 1, m_{t-1}^d = 1 \end{cases}$$

where $\delta_1^d = 0$. Write $\boldsymbol{\delta}_t$ for the vector of elapsed times at time stamp $t$ and $\Delta = \{\boldsymbol{\delta}_1, \boldsymbol{\delta}_2, ..., \boldsymbol{\delta}_T\}$.

The information available for patient $n$ is the triple $(\mathcal{X}_n, \mathcal{Y}_n, \mathcal{S}_n)$ The entire training set therefore is the sets of triples $\mathcal{D} = \{(\mathcal{X}_n, \mathcal{Y}_n, \mathcal{S}_n)\}_{n=1}^N$. We use functional notation to identify information about each patient, so $x_t^d(n)$ is the measurement of stream $d$ at time stamp $t$ for patient $n$, etc.

## 3.2 PROBLEM FORMULATION

At time stamp $T$, we have an array of measurements (which may or may not include the current label $y_T$); we must decide the next time $s_{T+1}$ at which to take new measurements and what measurements to conduct at that time. We measure the information provided by new samples by the effect on the label $y_{T+1}$, so we define the *predictive loss* from not sampling as the increase in uncertainty of our

prediction of $y_{T+1}$ and the *predictive gain* as the decrease in uncertainty of our prediction of $y_{T+1}$. Our approach is to find the first actual time $\tau$ at which the (estimated) predictive gain provided by new samples exceeds the cost of sampling (keeping in mind that the cost of sampling may be different for different streams), and to make the set of measurements at time $\tau$ that maximizes the (estimated) predictive gain minus the cost.

Our objective is to find the set of measurements that maximize (net) rewards, which we take as "information gain - cost"; this formulation is common; see Stachniss et al. (2005); Visser & Slamet (2008) for instance. Somewhat more formally, our objective is to solve the maximization problem:

$$\mathcal{C}_{T+1}^* = \arg \max_{\mathcal{C}_{T+1} \subset \mathcal{M}} \text{Information Gain}(\mathcal{C}_{T+1}) - \text{Cost}(\mathcal{C}_{T+1})$$

where $\mathcal{M}$ is the set of possible measurements. (For convenience, we assume here that the set of possible measurements is the same at every time but there would be no difficulty in allowing for the set of possible measurements to be different at different times. Note that the term "measurement" could actually encompass a panel of tests that can be made at the same cost.)

Solving this maximization problem presents two immediate problems. The first problem is that cost is well-defined and observable in our setting, but we need to decide the appropriate measure of information gain. Information gain is often defined Stachniss et al. (2005); Föllmer (1973) as the decrease in entropy. However, to properly compute the entropy, we should know the distribution of predictions. Instead, we use the decrease in uncertainty of prediction – measured as the difference between the upper and the lower bounds of the prediction – as our measure of information gain. The second problem is that maximizing over all possible subsets of measurements presents a potentially intractable problem; instead, we take a greedy approach that yields an approximation to the true optimum. We discuss both of these issues below.

The actual error in sampling stream $d$ at $s_{T+1} = \tau$ is the difference between the estimated values and the actual measurement $e_{T+1}^d = |\hat{x}_{T+1}^d - x_{T+1}^d|$. We don't know the actual error so we must construct an estimated error $\hat{e}_{T+1}^d$. Assuming that the distribution of errors is approximately normal (an assumption that is justified in the Appendix), the confidence intervals in the measurement of $x_{T+1}^d$ are of the form $CI_x = (\hat{x}_{T+1}^d - \lambda \hat{e}_{T+1}^d, \hat{x}_{T+1}^d + \lambda \hat{e}_{T+1}^d)$; e.g. $\lambda = 1.96$ for the 95% confidence level Rothenberg (1984); Davison & Hinkley (1997); Efron & Tibshirani (1986); Bartlett (1953). Note that the confidence intervals depend only on the estimates and not the true values (which are of course unknown). Each vector of estimates $(\hat{x}_{T+1}^d)$ of measurements, together with previous data (measured and inferred), can be used to produce a prediction $\hat{y}_{T+1}^d$ (see below). The confidence intervals for the stream measurements translate immediately into lower and upper confidence estimates $\hat{y}_{T+1}^{d,l}, \hat{y}_{T+1}^{d,u}$ (respectively) for the label prediction:

$$\hat{y}_{T+1}^{d,l} = \min_{\hat{x}_{T+1}^d \in CI_x} \hat{y}^d(\hat{x}_{T+1}^d, \mathcal{X}_T, \mathcal{S}_T) \qquad \hat{y}_{T+1}^{d,u} = \max_{\hat{x}_{T+1}^d \in CI_x} \hat{y}^d(\hat{x}_{T+1}^d, \mathcal{X}_T, \mathcal{S}_T) \qquad (1)$$

where $\mathcal{X}_T$ and $\mathcal{S}_T$ are previous measurements and measurement times until time stamp $T$. The (estimated) predictive gain in stream $d$ is therefore the difference $\hat{y}_{T+1}^{d,u} - \hat{y}_{T+1}^{d,l}$. Note that, because the minimization and maximization problems for each feature are independent each other, equation (1) can be solved by one-dimensional gradient descent.

Having defined gain, we now define (estimated) predictive gain minus cost as $F(\mathcal{C}_{T+1}, \mathcal{X}_T, \mathcal{S}_T)$. At each time $s_{T+1}$, we seek to find the subset $\mathcal{C}_{T+1} \subset \mathcal{M}$ of measurements that maximizes predictive gain minus cost; i.e. we wish to solve:

$$\mathcal{C}_{T+1}^* = \arg \max_{\mathcal{C}_{T+1} \subset \mathcal{M}} F(\mathcal{C}_{T+1}, \mathcal{X}_T, \mathcal{S}_T) \qquad (2)$$

However, if the number of possible measurements is large (which is typical), and there are complementarities among measurements, then solving the optimization problem (2) requires examining all possible subsets of measurements – which is intractable. Instead, we follow a greedy procedure: we identify all the individual streams $d$ with the property that the value of measuring that stream (by itself) exceeds the cost $c_d$ of sampling from that stream; we then take $C_{T+1}^*$ to be that set of measurements. This is a tractable optimization problem that yields an approximation to the actual optimal set of measurements. (We note again that every set of tests that can be carried out as a single panel at the same cost can be considered as a single test.) Hence we will actually solve the problem:

$$\mathcal{C}_{T+1}^* = \arg \max_{\mathcal{C}_{T+1} \subset \mathcal{M}} \sum_{d \in \mathcal{C}_{T+1}} \left( \hat{y}_{T+1}^{d,u}(\mathcal{X}_T, \mathcal{S}_T) - \hat{y}_{T+1}^{d,l}(\mathcal{X}_T, \mathcal{S}_T) - c^d \right) \qquad (3)$$

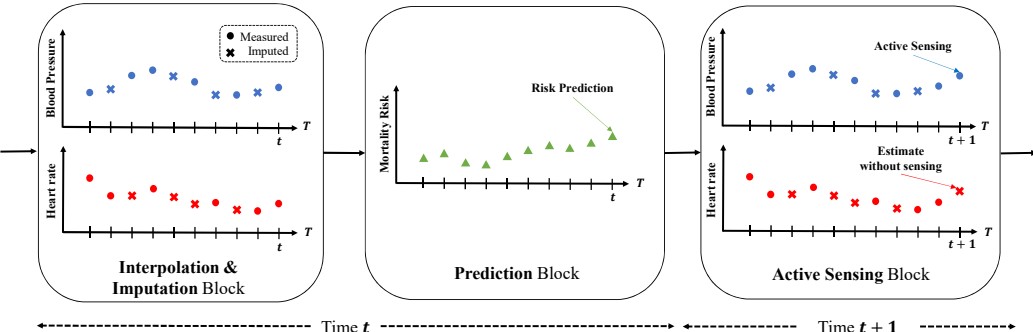

Figure 1: Deep Sensing Paradigm

It is important to note that $\mathcal{C}_{T+1}^*$ might be empty; i.e., there might be *no measurements* for which the information gain exceeds the cost. Because of this, Deep Sensing answers *both* the question "when to sample?" as well as the question "what to sample?" At each time $T$, Deep Sensing asks whether there are *any* measurements to be made at time $T+1$ for which the benefit outweighs the cost. If the answer is "yes" (i.e. $\mathcal{C}_{T+1}^* \neq \emptyset$) then Deep Sensing recommends that those measurements should be made at time $T+1$. If the answer is "no" (i.e. $\mathcal{C}_{T+1}^* = \emptyset$) then Deep Sensing asks whether there are any measurements to be made at time $T+2$ for which the benefit outweighs the cost, and so forth.

**Predicting Labels:** Given data (measured and inferred) until any time stamp $T'$, we generate a prediction $\hat{y}_{T'}$. The prediction rule can be learned from training data by any of various machine learning algorithms; we use a standard GRU-based RNN (Chung et al. (2014)). (See **Prediction** in Section 4.)

**Estimating the Values of New Measurements:** We view the problem of estimating new measurements as a special case of estimating missing measurements, so we begin by discussing our novel methods for this problem.

Fix data $\mathcal{D}$ through time stamp $T$. Assume that $x_t^d = *$. There are two standard methods to form an estimate $\hat{x}_t^d$: interpolation and imputation. *Interpolation* uses only the measurements $x_{t'}^d$ of the fixed data stream $d$ for other time stamps $t'$ (perhaps both before and after $t$). Interpolation ignores the correlation with other data streams. *Imputation* uses only the measurements $x_t^{d'}$ at the fixed time $t$ for other data streams $d'$. Imputation ignores the correlation with other times.

In principle, we could try to form the estimate $\hat{x}_t^d$ by using all the information in $\mathcal{D}$. However, this would require learning a vast number of parameters and hence a vast number of training instances, so this is impractical. Instead, we propose an efficient hierarchical learning framework using a novel RNN architecture that allows us to capture the correlations both *within streams* and *across streams*. The entire process of Deep Sensing is illustrated in Fig. 1.

## 4 DEEP SENSING: ALGORITHM

In this section, we describe the training and runtime stages of the Deep Sensing algorithm. Fig. 2 shows block diagrams of the two stages.

### 4.1 TRAINING STAGE

The training stage of Fig. 2 shows the block diagram of the entire 5 blocks of the training stage. The first four blocks train the Interpolation block ($\boldsymbol{\Phi}$), the Imputation block ($\boldsymbol{\Psi}$), the Error Estimation block ($\boldsymbol{\Gamma}$) and the Prediction block ($\boldsymbol{\Omega}$). The Adaptive Sampling block creates multiple presentations based on different sets of missing data. (The importance of this will be explained below.)

**Loss function:** The objective of the interpolation and imputation blocks is to minimize the error that would be made in estimating missing measurements. Evidently, we cannot estimate the error of

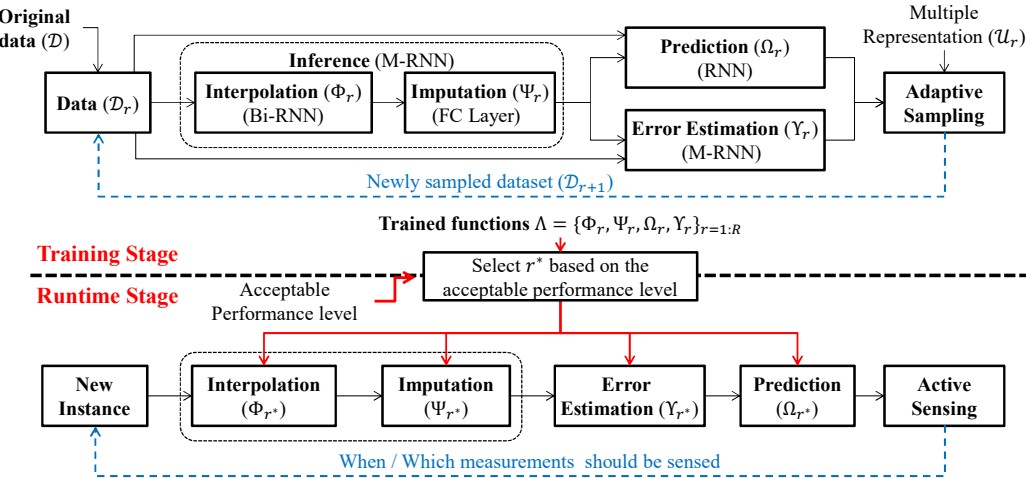

Figure 2: Block diagram of Deep Sensing

a measurement that is truly missing in the dataset. Instead we fix a measurement that was actually made, remove that measurement, form an estimate for the measurement using only the data set $\mathcal{D} - x_t^d$ (i.e. the data set with $x_t^d$ removed), and then compute the error between the estimate and the actual measurement (that was deleted). If $x_t^d$ is an actual measurement and $\hat{x}_t^d$ is the estimate formed when $x_t^d$ is removed then the loss can be defined as the mean squared error (MSE) $l(\hat{x}_t^d, x_t^d) = (\hat{x}_t^d - x_t^d)^2$. The loss for the entire dataset $\mathcal{D}$ is defined as

$$\mathcal{L}(\{\hat{x}_t^d, x_t^d\}) = \sum_{n=1}^{N} \left[ \frac{\sum_{t=1}^{T_n} \sum_{d=1}^{D} m_t^d(n) \times (\hat{x}_t^d(n) - x_t^d(n))^2}{\sum_{t=1}^{T_n} \sum_{d=1}^{D} m_t^d(n)} \right].$$

**Interpolation:** The objective of the interpolation block is to construct an interpolation function $\mathbf{\Phi}$ that operates *within* a stream. To emphasize that the estimate for $x_t^d$ depends on the data with $x_t^d$ removed, we abuse notation and write $\tilde{x}_t^d = \mathbf{\Phi}(\mathcal{D} - x_t^d)$. (Keep in mind that we are actually only using the data from stream $d$, not the data from other streams.) We construct the estimation function $\mathbf{\Phi}$ using a bi-directional recurrent neural network (Bi-RNN) with a Gated Recurrent Unit (GRU). However, unlike a conventional Bi-RNN (Graves & Schmidhuber (2005)), the timing of inputs into the hidden layer is lagged in the forward direction and advanced in the backward direction: at time $t$, inputs of forward hidden states come from $t - 1$ and inputs of backward hidden states come from $t + 1$. Mathematically, we have:

$$\mathbf{o}_t = \overrightarrow{W} \overrightarrow{\mathbf{h}}_t + \overleftarrow{W} \overleftarrow{\mathbf{h}}_t + \mathbf{c}_o,$$

$$\overrightarrow{\mathbf{h}}_t = (1 - \overrightarrow{\mathbf{z}}_t) \odot \overrightarrow{\mathbf{h}}_{t-1} + \overrightarrow{\mathbf{z}}_t \odot \overrightarrow{\tilde{\mathbf{h}}}_t, \qquad \overleftarrow{\mathbf{h}}_t = (1 - \overleftarrow{\mathbf{z}}_t) \odot \overleftarrow{\mathbf{h}}_{t+1} + \overleftarrow{\mathbf{z}}_t \odot \overleftarrow{\tilde{\mathbf{h}}}_t,$$

$$\overrightarrow{\mathbf{z}}_t = \sigma(\overrightarrow{W}_z \mathbf{x}_{t-1} + \overrightarrow{U}_z \overrightarrow{\mathbf{h}}_{t-1} + \overrightarrow{V}_z \boldsymbol{\delta}_{t-1} + \overrightarrow{\mathbf{c}}_z), \qquad \overleftarrow{\mathbf{z}}_t = \sigma(\overleftarrow{W}_z \mathbf{x}_{t+1} + \overleftarrow{U}_z \overleftarrow{\mathbf{h}}_{t+1} + \overleftarrow{V}_z \boldsymbol{\delta}_{t+1} + \overleftarrow{\mathbf{c}}_z),$$

$$\overrightarrow{\tilde{\mathbf{h}}}_t = \phi(\overrightarrow{W}_h \mathbf{x}_{t-1} + \overrightarrow{U}_h (\overrightarrow{\mathbf{r}}_t \odot \overrightarrow{\mathbf{h}}_{t-1}) + \overrightarrow{V}_h \boldsymbol{\delta}_{t-1} + \overrightarrow{\mathbf{c}}_h),$$

$$\overleftarrow{\tilde{\mathbf{h}}}_t = \phi(\overleftarrow{W}_h \mathbf{x}_{t+1} + \overleftarrow{U}_h (\overleftarrow{\mathbf{r}}_t \odot \overleftarrow{\mathbf{h}}_{t+1}) + \overleftarrow{V}_h \boldsymbol{\delta}_{t+1} + \overleftarrow{\mathbf{c}}_h),$$

$$\overrightarrow{\mathbf{r}}_t = \sigma(\overrightarrow{W}_r \mathbf{x}_{t-1} + \overrightarrow{U}_r \overrightarrow{\mathbf{h}}_{t-1} + \overrightarrow{V}_r \boldsymbol{\delta}_{t-1} + \overrightarrow{\mathbf{c}}_r), \qquad \overleftarrow{\mathbf{r}}_t = \sigma(\overleftarrow{W}_r \mathbf{x}_{t+1} + \overleftarrow{U}_r \overleftarrow{\mathbf{h}}_{t+1} + \overleftarrow{V}_r \boldsymbol{\delta}_{t+1} + \overleftarrow{\mathbf{c}}_r)$$

where $\odot$ is element-wise multiplication, $\sigma$ is the sigmoid function, $\phi$ is $\tanh$ function, and arrows indicate forward/backward direction. The output $\mathbf{o}_t$ is the interpolated value $\tilde{\mathbf{x}}_t$. In this interpolation block, we are only using/capturing the temporal correlation within each stream. As a consequence, the matrices $U, V, W$ are block-diagonal. Hence the total number of parameters that must be learned is on the order of the number $D$ of streams. (Recall that in a standard Bi-RNN, the number of parameters to be learned will be on the order of the square $D^2$ of the number of streams.) This

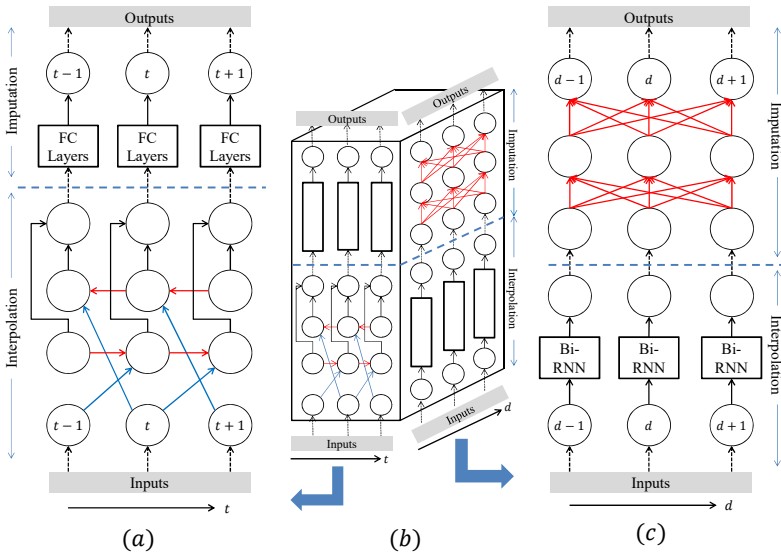

Figure 3: Diagram of the neural networks for M-RNN

avoids overfitting and leads to significant performance improvements as compared to a standard Bi-RNN. (See the Interpolation part of Fig. 3.)

**Imputation:** The objective of the imputation block is to construct an imputation function $\boldsymbol{\Psi}$ that operates *across* streams. Again, we abuse notation and write $\check{x}_t^d = \boldsymbol{\Psi}(\mathcal{D} - x_t^d)$. Keep in mind that now we are using only data at time stamp $t$, not data from other time stamps. We construct the function $\boldsymbol{\Psi}$ to be independent of the time stamp $t$; so we construct it using fully connected layers (FC); see Imputation part of Fig 3:

$$\mathbf{o}_t = W\mathbf{h}_t + \mathbf{c}_o,$$
$$\mathbf{h}_t = U\mathbf{x}_t + V\tilde{\mathbf{x}}_t + Q\mathbf{m}_t + \mathbf{c}_h$$

where $\mathbf{o}_t = \hat{\mathbf{x}}_t$ and the block-diagonal entries of $U$ are zero because we do not use $x_t^d$ to estimate $\hat{x}_t^d$. We use multiple deeply stacked FC layers using linear activation functions.

We jointly learn the functions $\boldsymbol{\Phi}$ and $\boldsymbol{\Psi}$ using the stacked networks of Bi-RNN and FC layers.

$$\boldsymbol{\Phi}^*, \boldsymbol{\Psi}^* = \arg\min_{\boldsymbol{\Phi}, \boldsymbol{\Psi}} \left[ \mathcal{L}(\{\boldsymbol{\Psi}\Big(\{x_t^d, \boldsymbol{\Phi}\Big(\{x_t^d, m_t^d, \delta_t^d\}_{t=1:T}\Big), m_t^d\}_{d=1:D}\Big), x_t^d\}) \right]$$

We refer to the entire structure as a *Multi-directional Recurrent Neural Network (M-RNN)*; see Fig.3.

**Prediction:** Now that we have a procedure to reconstruct (interpolate/impute) missing data, we use the reconstructed data to predict the labels. We accomplish this in the prediction block. Note that for prediction, we use actual measurements when available and estimated measurements when actual measurements are missing (not available). We also use as an input the *mask vector* (the indicator of missingness), which provides to the network the information as to whether measurements are actual or estimated. Once again, we construct the function to minimize the prediction error when we predict an observed label. The loss function is defined as $\mathcal{L}(\{\hat{y}_t, y_t\}) = \frac{1}{N}\sum_{n=1}^{N}\frac{\sum_{t=1}^{T_n}(\hat{y}_t(n)-y_t(n))^2}{T_n}$ The prediction block optimizes the function:

$$\boldsymbol{\Omega}^* = \arg\min_{\boldsymbol{\Omega}} \left[ \mathcal{L}(\{\hat{y}_t, y_t\}) \right] = \arg\min_{\boldsymbol{\Omega}} \left[ \mathcal{L}(\{\boldsymbol{\Omega}\Big(\{x_t^d, \hat{x}_t^d, \delta_t^d\}_{t=1:T,d=1:D}\Big), y_t\}) \right]$$

Note that we use the time intervals $\delta_t^d$ as inputs to the prediction function in order to deal with the fact that the data streams are irregularly sampled. This optimization problem is a standard problem for timely prediction so we can use a standard GRU-based RNN (Chung et al. (2014)) to solve it.

**Error Estimation:** At runtime, we have to decide when/what to sample in the active sensing block. We make this decision on the basis of predictive gain which is determined by the difference between our estimate of a measurement and what the actual value of the measurement would be; the actual error $e_t^d = |\hat{x}_t^d - x_t^d|$. Of course, we do not know what this will be because we do not know what the actual value of the measurement would be. Hence we need an *estimate* $\hat{e}_t^d$ of the actual error. As before, we construct this estimate on the basis of the actual training data that we have. For tractability, we posit that this estimate depends on the pattern of missing data and on time intervals of the measurements but not on actual measurements. We use the same mean square loss function:

$$\mathcal{L}(\{\hat{e}_t, e_t\}) = \sum_{n=1}^{N} \frac{\sum_{t=1}^{T_n} \sum_{d=1}^{D} m_t^d(n) \times (\hat{e}_t^d(n) - e_t^d(n))^2}{\sum_{t=1}^{T_n} \sum_{d=1}^{D} m_t^d(n)}.$$

Hence we need to solve for the function

$$\mathbf{\Gamma}^* = \arg\min_{\mathbf{\Gamma}} \left[ \mathcal{L}(\{\hat{e}_t^d, e_t^d\}) \right] = \arg\min_{\mathbf{\Gamma}} \left[ \mathcal{L}(\{\mathbf{\Gamma}\left( \{m_\tau^d\}_{d=1:D}, \{\delta_t^d, m_t^d\}_{t=1:T} \right), e_t^d\}) \right].$$

Because this involves both inter-stream and intra-stream variables, we again use our M-RNN structure. However, the inputs and outputs are different: for the interpolation and imputation blocks, the inputs are the measurements $\{x_\tau^d\}$, sensing information $\{m_\tau^d\}$ and the time intervals $\{\delta_t^d\}$ and the output is the estimated measurement $\hat{x}_t^{d'}$. For the error estimation block, the inputs are the sensing information $\{m_\tau^d\}$ and the time intervals $\{\delta_t^d\}$ and the output is the estimation error $\hat{e}_t^d$.

It is useful to understand the relationship between the mask vector (which indicates missing measurements) and the three different categories of missingness of measurements. (1) If the measurements in the dataset are Missing Completely At Random (MCAR) then the mask vector is independent of the observable features/measurements. (2) If the measurements in the dataset are Missing At Random (MAR) then the correlation between the mask vector can be completely explained by the observable features/measurements. (3) If the measurements in the dataset are Missing Not At Random (MNAR) then the mask vector (and the values of the missing features) cannot be completely explained by the observable features/measurements.

We have focused on the MCAR and MAR settings because the values of missing measurements are estimated based on the observable features. However, our approach also has something useful to say in the MNAR setting as well, because we use the mask vector – which depends on both observed and unobserved variables and therefore incorporates "informative missingness" – as an input for both estimation of missing values and for prediction. We demonstrate this point in Section 5.4.

**Adaptive Sampling:** As pointed out in Section 3, the decision of what/when to sense arises from trading off the cost of measurement against the predictive gain of measurement. To this point, we have constructed a procedure that achieves a certain performance – predictive gain – at a prescribed cost. If we are willing to settle for a lower level of performance, we can do so at lower cost by sampling less often. To know how much less often to sample we need to know how much information would be lost if we sampled less often, which we can determine by carrying out the previous procedure to produce different presentations, each based on a different pattern of missing data. For each presentation we need to train the functions $\mathbf{\Phi}, \mathbf{\Psi}, \mathbf{\Omega}, \mathbf{\Gamma}$ on the appropriate training set, which is smaller than the original training set.

To construct these presentations, we begin with the original training set and remove additional measurements. We should not do this at random, but rather using the informational criteria we use to decide on active sensing at runtime: remove measurements whose predictive gain is below a given threshold. We call this *adaptive sampling*. This will yield a decreasing sequence of data sets $\mathcal{D}_0 \supset \mathcal{D}_1 \supset ... \supset \mathcal{D}_R$ (where $\mathcal{D}_0 = \mathcal{D}$, the original dataset).

The training procedures for the functions $\mathbf{\Phi}, \mathbf{\Psi}, \mathbf{\Omega}, \mathbf{\Gamma}$ are as follows. Fix thresholds $u_1, \ldots, u_R > 0$. (In practice, these would be specified by the user.) We begin with $\mathcal{D}_0$. For each measurement $x_t^d(n) \in \mathcal{D}_0$ we use the current functions $\mathbf{\Phi} = \mathbf{\Phi}_0, \mathbf{\Psi} = \mathbf{\Psi}_0, \mathbf{\Omega} = \mathbf{\Omega}_0, \mathbf{\Gamma} = \mathbf{\Gamma}_0$ to compute the predictive gain from that measurement in the current dataset. We sequentially delete all the measurements whose predictive gain ("information gain - cost") is below the prescribed threshold $u_1$; this yields a resampled data set $\mathcal{D}_1$. We now retrain on $\mathcal{D}_1$ to obtain new functions $\mathbf{\Phi}_1, \mathbf{\Psi}_1, \mathbf{\Omega}_1, \mathbf{\Gamma}_1$ and repeat the same procedure: for each measurement $x_t^d(n) \in \mathcal{D}_1$, we compute the predictive gain from

that measurement in the current dataset and sequentially delete those measurements whose predictive gain is below threshold $u_2$, etc. We repeat the above procedures continuing through whatever set of thresholds are chosen. (It is important to keep in mind that, in the active sampling process, if the actual dataset is not complete, we only consider measurements that are actually recorded in the dataset. Thus we are never confronted with the need to compare an estimate/prediction against data that is actually missing.)

Note that if we increase the threshold $u_r$, we delete more data and retain fewer samples, so the expenditure on sampling is smaller. However because we have trained on fewer samples, our predictions will be less accurate. This creates the cost-performance trade-off. Fig. 2 illustrates the entire block diagram of Deep Sensing. Fig 5 in the Appendix details the operation of Deep Sensing in runtime. Pseudo-codes of Deep Sensing for the training and runtime stages are shown in the Appendix.

## 5 EXPERIMENTS

In this section, we evaluate the performance of Deep Sensing using two real-world medical datasets. Our experimental results present three sets of comparisons: active sensing, prediction, and missing value inference. The first comparison shows the performance gain of Deep Sensing (in comparison with benchmarks) in sensing the critical measurements for prediction. The second and third comparisons show the performance gain of the M-RNN algorithm in estimating missing values and the effect on prediction accuracy. We describe all configurations of the various algorithms in the Appendix.

### 5.1 DATA DESCRIPTION

We conducted our experiments using two real-world medical datasets. The first of these datasets is MIMIC-III (Johnson et al. (2016)) which records data on patients in intensive care units (ICU). We used 22,803 patients who admitted were to ICU after 2008. We use the 20 vital signs which were most frequently measured and for which missing rates are lowest (e.g. heart rate, respiratory rate) and 20 lab tests (e.g. creatinine, chloride). Thus we have 40 physiological data streams in all. Vital signs were taken approximately every 1 hour; lab tests were taken approximately every 24 hours. For this dataset, the adverse event we predict is death. The second of these datasets, which we call Wards, was assembled and described by Alaa et al. (2017b). (We are grateful to the authors for sharing this dataset with us.) The Wards dataset records 37 physiological data streams (vital signs and lab tests) on 6,094 patients who were hospitalized in a major medical center in 2013-2015. Vital signs were taken approximately every 4 hours; lab tests were taken approximately every 24 hours. For this dataset, the adverse event we predict is admission to ICU as a result of clinical deterioration.

### 5.2 SIMULATION SETUP

We randomly divided the dataset into a mutually exclusive training set (80%) and testing set (20%). We conducted 10 independent experiments with different combinations of training/testing sets; we report the mean and variance of the performance in the 10 experiments. In our experiments we are trying to predict which patients will experience an adverse event (death for the MIMIC-III dataset and ICU admission for the Wards dataset) within 24 hours from the current time. At each time, we assign the label 1 to patients who experienced the relevant adverse event within 24 hours; for other patients we assigned the label 0. (Formally: $y_t = 1$ for $s_t > S_T - 24$ and $y_t = 0$ for $s_t \leq S_T - 24$ where $S_T$ is the time that the adverse event occurred. Of course the true label $y_t$ is not observed at time stamp $s_t$.)

**Active sensing:** To evaluate the performance of Deep Sensing, we graph predictive accuracy – area under the ROC curve (AUC) – against cost. (The cost of each possible measurement is well-defined in the medical domain. If all measurements were equally costly, we could simply identify the cost with the observation rate. Because some measurements are most costly than others, we simply weight those measurements more heavily when expressing the cost in terms of the observation rate. In this case, we take the cost for lab tests to be 5 times the cost for taking vital signs so weight lab tests accordingly.) We compare the predictive accuracy of Deep Sensing with multiple presentations, Deep Sensing with a single presentation (using only the original dataset to train), Deep Sensing with

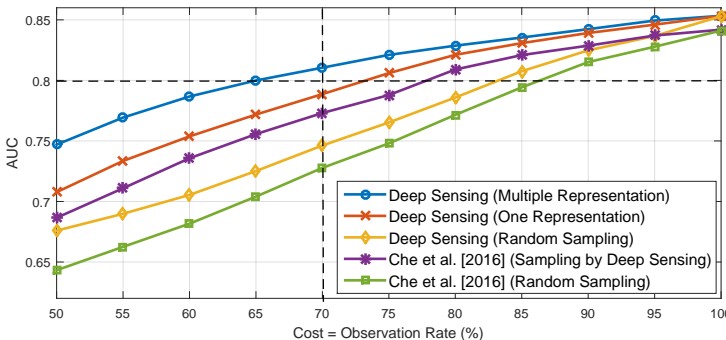

Figure 4: Active Sensing: AUC vs Cost for Different Solutions with MIMIC-III dataset (Lab test cost $= 5\times$ Vital sign cost)

random sampling, and two benchmarks based on the method of Che et al. (2016) for prediction with missing data (sampling either using the method of Deep Sensing or randomly).

**Prediction:** We also evaluate prediction given only the available observations. The prediction we consider is the adverse event (death or clinical deterioration); we use AUC as the performance metric. We compare the performance of Deep Sensing with four state-of-the-art RNN timely-prediction models and a GRU-based RNN method using conventional estimation methods for interpolation and imputation. To make the comparison fair, we use GRU-based RNNs for each benchmark. Deep Sensing is compared with the benchmarks in two settings. In setting A, we sampled 60% of the measurements; for Deep Sensing, we used the Deep Sensing algorithm, for the benchmarks, we use random sampling. In setting B we sampled 60% of the measurements, using the Deep Sensing algorithm everywhere.

**Estimation of missing values:** To evaluate the performance of the M-RNN algorithm (the combination of interpolation and imputation block) in estimating missing values, we compare with other standard methods: interpolations (Spline and Cubic Kreindler & Lumsden (2012)), imputations (MICE (White et al. (2011)), Kernel (Rehfeld et al. (2011)) and EM (García-Laencina et al. (2010))). We randomly remove 30% of the observations and treat them as missing. We then estimate the missing observations values using our M-RNN algorithm and benchmarks. We use the root mean square error (RMSE) between estimated values and actual observed values as the performance metric.

**Runtime of Deep Sensing:** Deep Sensing is computationally efficient. For instance, on the MIMIC-III dataset (23,200 samples, 40 dimensions, 25 time stamps), Deep Sensing takes less than 1 hour on a machine with i7-6900K CPU (3.2GHz x 16) and 64GB RAM. By comparison, the most common imputation method, MICE (implemented in R packages) takes 11 hours on the same machine.

### 5.3 SIMULATION RESULTS AND DISCUSSION

**Active Sensing:** As Fig. 4 illustrates, Deep Sensing predicts best for every specification of cost; equivalently, Deep Sensing expends the least cost for every specified prediction accuracy. Fig. 4 also shows that the performance gains achieved by Deep Sensing come *both* from active sampling and from better inference. As seen in Figure 4, if the observation rate were 100% (so there would be no gain from active sensing), the AUC improvement would be limited. However, as the observation rate decreases the AUC gain increases because Deep Sensing actively decides what to sample and when to sample, thereby providing results that are much superior to random sampling.

**Prediction:** Tables 1 and 2 provide the mean, standard deviation, and performance gain (%) (in terms of AUC) from Deep Sensing in comparison to the benchmarks for two real-world medical datasets. Table 1 and 2 show that Deep Sensing provides significant gains of the prediction accuracy for both datasets (around 30% in Setting A and 20% in Setting B for all the benchmarks). The significant gains for prediction come from the combination of accurate missing value inference and active sensing as seen in Figure 4.

Table 1: AUC for Deep Sensing and Benchmarks with MIMIC-III dataset (See the text for descriptions of Settings A, B). *: p-value $< 0.05$

| AUC (Mean $\pm$ Std (Gain %)) | | **MIMIC-III** (Setting A) | **MIMIC-III** (Setting B) |
|---|---|---|---|
| **Proposed Model** | **Deep Sensing** | **0.8019 $\pm$ 0.0112 (-)** | **0.8019 $\pm$ 0.0112 (-)** |
| RNN based | Choi et al. (2015)* | 0.7112 $\pm$ 0.0134 (31.4 %) | 0.7598 $\pm$ 0.0110 (17.5 %) |
| | Lipton et al. (2016)* | 0.7072 $\pm$ 0.0108 (32.3 %) | 0.7551 $\pm$ 0.0115 (19.1 %) |
| | Che et al. (2016)* | 0.7133 $\pm$ 0.0111 (30.9 %) | 0.7593 $\pm$ 0.0123 (17.7 %) |
| | Futoma et al. (2017)* | 0.7094 $\pm$ 0.0121 (31.8 %) | 0.7579 $\pm$ 0.0129 (18.2 %) |
| Interpolation + RNN | Spline* | 0.7045 $\pm$ 0.0137 (33.0 %) | 0.7542 $\pm$ 0.0108 (19.4 %) |
| | Cubic* | 0.7012 $\pm$ 0.0129 (33.7 %) | 0.7569 $\pm$ 0.0112 (18.5 %) |
| Imputation + RNN | MICE* | 0.7093 $\pm$ 0.0132 (31.9 %) | 0.7571 $\pm$ 0.0121 (18.4 %) |
| | Kernel* | 0.7002 $\pm$ 0.0119 (33.9 %) | 0.7534 $\pm$ 0.0139 (19.7 %) |
| | EM* | 0.7019 $\pm$ 0.0098 (33.5 %) | 0.7531 $\pm$ 0.0107 (19.8 %) |

Table 2: AUC for Deep Sensing and Benchmarks with Wards dataset (See the text for descriptions of Settings A, B). *: p-value $< 0.05$

| AUC (Mean $\pm$ Std (Gain %)) | | **Wards** (Setting A) | **Wards** (Setting B) |
|---|---|---|---|
| **Proposed Model** | **Deep Sensing** | **0.8348 $\pm$ 0.0201 (-)** | **0.8348 $\pm$ 0.0201 (-)** |
| RNN based | Choi et al. (2015)* | 0.7739 $\pm$ 0.0264 (26.9 %) | 0.8028 $\pm$ 0.0184 (16.2 %) |
| | Lipton et al. (2016)* | 0.7893 $\pm$ 0.0237 (21.6 %) | 0.8107 $\pm$ 0.0191 (12.7%) |
| | Che et al. (2016)* | 0.7905 $\pm$ 0.0143 (21.1 %) | 0.8159 $\pm$ 0.0160 (10.3 %) |
| | Futoma et al. (2017)* | 0.7911 $\pm$ 0.0193 (20.9 %) | 0.8177 $\pm$ 0.0147 (9.4 %) |
| Interpolation + RNN | Spline* | 0.7829 $\pm$ 0.0085 (23.9 %) | 0.8023 $\pm$ 0.0187 (16.4 %) |
| | Cubic* | 0.7712 $\pm$ 0.0084 (27.8 %) | 0.7993 $\pm$ 0.0137 (17.7%) |
| Imputation + RNN | MICE* | 0.7499 $\pm$ 0.0096 (33.9%) | 0.7877 $\pm$ 0.0149 (22.2 %) |
| | Kernel* | 0.7397 $\pm$ 0.0155 (36.5 %) | 0.7728 $\pm$ 0.0187 (27.3 %) |
| | EM* | 0.7593 $\pm$ 0.0168 (31.4%) | 0.7784 $\pm$ 0.0163 (25.5%) |

**Estimation of missing values:** Table 3 shows the mean and standard deviation of the RMSE for M-RNN and benchmarks for both the MIMIC-III and the Wards dataset. The RMSE of M-RNN is less than half that of the best benchmark in MIMIC-III dataset and less than 70% that of the best benchmark in Wards dataset. All the improvements are statistically significant (p-value $< 0.05$).

Table 3: RMSE of Missing information for M-RNN and Benchmarks with MIMIC-III and Wards datasets. *: p-value $< 0.05$

| Datasets | | | | **Interpolation** | | **Imputation** | | |
|---|---|---|---|---|---|---|---|---|
| | **Metrics** | **M-RNN** | **Spline** | **Cubic** | **MICE** | **Kernel** | **EM** | |
| **MIMIC-III** | RMSE - Mean | **0.0137** | 0.0735* | 0.0279* | 0.0611* | 0.0556* | 0.0467* | |
| | RMSE - Std | **0.0013** | 0.0012 | 0.0013 | 0.0011 | 0.0011 | 0.0014 | |
| **Wards** | RMSE - Mean | **0.0169** | 0.0314* | 0.0211* | 0.0554* | 0.0627* | 0.0761* | |
| | RMSE - Std | **0.0019** | 0.0011 | 0.0021 | 0.0013 | 0.0014 | 0.0017 | |

## 5.4 SOURCE OF GAINS

Our M-RNN architecture consists of two components: the interpolation block (with forward/backward connection) and the imputation block). To understand the source of gains provided by the various components of our approach, we carry out a series of experiments in which we restrict

the operation of our architecture in various ways. In the first experiment, we restrict to interpolation only (no imputation), in the second experiment, we restrict to imputation only (no interpolation), in the third experiment we restrict to forward interpolation only (no backward interpolation), and in the fourth experiment we replace the GRU based RNN with a standard RNN. Table 4 provides the results of these experiments. As can be seen, both the interpolation and imputation blocks by themselves provide significant performance improvements. (Because the sampling frequencies are high in both datasets, the performance gain of the interpolation block is higher than that of the imputation block.) The backward connection also improves performance, but only marginally (approximately 10%). Finally, using the GRU-based RNN significantly improves performance by capturing long-term dependencies in an efficient way.

As we have discussed above, and as is illustrated in Section 4, Deep Sensing can be applied if data is missing completely at random (MCAR) or missing at random (MAR), but it can also be applied if data is missing not at random (MNAR).[1] To apply Deep Sensing in the MNAR setting, we incorporate the mask vector (the indicator of the missingness) as an additional input to capture "informative missingnes". As can be seen in the last row in Table 4, doing so leads to a significant improvement – approximately 30%.

Table 4: Source of gain analysis for M-RNN with MIMIC-III and Wards datasets. (Performance metric: RMSE)

| Benchmarks | MIMIC-III | | Wards | |
|---|---|---|---|---|
| | Mean ± Std | Loss (%) | Mean ± Std | Loss (%) |
| M-RNN | 0.0137 ± 0.0013 | (-) | 0.0169 ± 0.0019 | (-) |
| No Imputation | 0.0188 ± 0.0011 | 27.1% | 0.0201 ± 0.0016 | 15.9% |
| No Interpolation | 0.0285 ± 0.0018 | 51.9% | 0.0278 ± 0.0024 | 39.2% |
| No Backward Interpolation | 0.0151 ± 0.0014 | 9.3% | 0.0179 ± 0.0018 | 5.6% |
| Standard RNN | 0.0178 ± 0.0011 | 23.0% | 0.0194 ± 0.0016 | 12.9% |
| Without Mask Vector | 0.0226 ± 0.0021 | 39.4% | 0.0247 ± 0.0027 | 31.6% |

## 6 CONCLUSION

The problem of active sensing is a very important one but has not been thoroughly treated in the literature. We present here a solution based on a novel deep learning architecture. As part of the solution, we provide a new method for reconstructing missing data that exploits joint interpolation within data streams and imputation across data streams. We demonstrate that Deep Sensing makes large and statistically significant improvements in comparison with state-of-the-art benchmarks in two real-world datasets.

## ACKNOWLEDGMENTS

This work was supported by the Office of Naval Research (ONR) and the NSF (Grant number: ECCS1462245, ECCS1533983, and ECCS1407712).

---

[1] This setting is obviously important, but the literature dealing with it is small; see Alaa et al. (2017a).

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

APPENDIX

JUSTIFICATION OF THE APPROXIMATE CONFIDENCE INTERVAL IN SECTION 3

Define $\hat{x} = x + n$ where $x$ is a measurement and $\hat{x}$ is the estimated measurement. For the moment, assume that $n$ is Gaussian noise and that we can interpret the error $e = |\hat{x} - x|$ as the estimated standard deviation of the Gaussian noise $n$. Then, $(\hat{x} - \lambda e, \hat{x} + \lambda e)$ is the proper confidence interval for x in the formal statistical sense.

The assumption of Gaussian noise is quite standard and probably needs no further comment. The interpretation of the error ($|\hat{x} - x|$) as the estimated standard deviation of Gaussian noise is not standard but can be justified in the following way. Let us assume that our measurement $x$ comes from a Gaussian distribution $\mathcal{N}(\mu, \sigma^2)$. If our estimate $\hat{x}$ is the expected value of $x$ (i.e. $\mu$), then we will have $x = \hat{x} + n$, where $x$ is the observed measurement from a Gaussian distribution, $\hat{x}$ is the expected value of $x$ and $n$ is normal (Gaussian) distribution $\mathcal{N}(0, \sigma^2)$. In that case, the expected value of $e = |\hat{x} - x| = |n|$ is just the standard deviation of Gaussian noise, which is $\sqrt{\mathbb{E}[n^2]} = \sigma$. Hence, we need two assumptions: (1) our estimate $\hat{x}$ is the expected value of $x$; (2) the observed measurement can be approximated as the sum of the expectation of $x$ and Gaussian noise (approximate normality [Rothenberg (1984); Davison & Hinkley (1997); Efron & Tibshirani (1986); Bartlett (1953)]).

More formally, assume that the measurement $x$ is sampled from an unknown distribution $\mathcal{P}_\theta$; i.e. $x \sim \mathcal{P}_\theta$. If $\mathcal{P}_\theta$ is itself normal (Gaussian), then it follows that.

$$x \sim P_\theta \Leftrightarrow x = \mathbb{E}[x] + n$$

where $n \sim \mathcal{N}(0, \sigma^2)$. (This uses the observation that the expectation of the normal distribution is the mean). In general, we cannot assume that $\mathcal{P}_\theta$ is normal, but it will be enough if it is approximately normal, which is a common assumption in the literature (see [Rothenberg (1984); Davison & Hinkley (1997); Efron & Tibshirani (1986); Bartlett (1953)] for instance.) In that case, following the literature we can obtain

$$x \simeq \mathbb{E}[x] + n$$

where $n \sim \mathcal{N}(0, \mathbb{E}[(x - \mathbb{E}[x])^2])$. From this we obtain that $\hat{x} = \mathbb{E}[x]$. (In practice, we use $\hat{x}$ as the sample mean of $x$ which converges to $\mathbb{E}[x]$). In that case the distribution of the error $e = |\hat{x}x|$ coincides with the distribution of the absolute value of samples generated by the normally distributed noise $n$:

$$\mathbb{E}[e] = \mathbb{E}[|\hat{x} - x|] = \mathbb{E}[\sqrt{(\hat{x} - x)^2}] = \mathbb{E}[\sqrt{n^2}] = \mathbb{E}[\sqrt{(x - \mathbb{E}[x])^2}].$$

Thus, estimating $e$ can be interpreted as estimating the standard deviation of the measurements $x$.

THE OPERATION OF DEEP SENSING IN RUNTIME

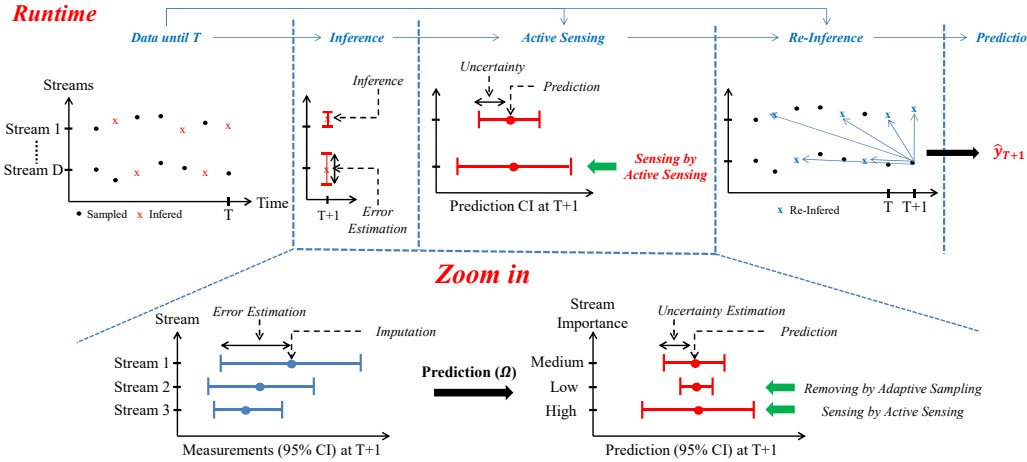

Figure 5: The operation of Deep Sensing in runtime

PSEUDO-CODES OF DEEP SENSING

---
**Algorithm 1** Deep Sensing - Training Stage
---

**Input:** Dataset $\mathcal{D} = \{(\mathcal{X}_n, \mathcal{Y}_n, \mathcal{S}_n)\}_{n=1}^N$, multiple representations $\mathcal{U} = \{u_1, u_2, ..., u_R\}$
**For** each representation $u_r \in \mathcal{U}$
    **Initialization:**
        $\mathbf{\Phi}_1, \mathbf{\Psi}_1, \mathbf{\Omega}_1, \mathbf{\Gamma}_1 \leftarrow$ Xavier Initialization, $\mathcal{D}_1 \leftarrow \mathcal{D}$
        $\mathbf{\Phi}_r \leftarrow \mathbf{\Phi}_{r-1}, \mathbf{\Psi}_r \leftarrow \mathbf{\Psi}_{r-1}, \mathbf{\Omega}_r \leftarrow \mathbf{\Omega}_{r-1}, \mathbf{\Gamma}_r \leftarrow \mathbf{\Gamma}_{r-1}$ and $D_r \leftarrow D_{r-1}$
    **Interpolation and Imputation:** Using M-RNN and FC layers

$$\mathbf{\Phi}_r^*, \mathbf{\Psi}_r^* = \arg\min_{\mathbf{\Phi},\mathbf{\Psi}} \left[ \mathcal{L}(\{\mathbf{\Phi}\Big(\{x_\tau^d, \mathbf{\Psi}\Big(\{x_\tau^d, m_\tau^d, \delta_\tau^d\}_{\tau=1:T}\Big), m_\tau^d\}_{d=1:D}\Big), x_\tau^d\}) \right]$$

    **Error Estimation:** Using M-RNN and FC layers

$$\mathbf{\Gamma}_r^* = \arg\min_{\mathbf{\Gamma}} \left[ \mathcal{L}(\{\mathbf{\Gamma}\Big(\{m_\tau^d\}_{d=1:D}, \{\delta_\tau^d, m_\tau^d\}_{\tau=1:T}\Big), e_\tau^d\}) \right]$$

    **Prediction:** Using RNN with GRU

$$\mathbf{\Omega}_r^* = \arg\min_{\mathbf{\Omega}} \left[ \mathcal{L}(\{\mathbf{\Omega}\Big(\{x_\tau^d, \hat{x}_\tau^d, \delta_\tau^d\}_{\tau=1:T,d=1:D}\Big), y_\tau\}) \right]$$

    **Adaptive Sampling:**
        **For** all $\tau, s, n$
            **If** $\hat{y}_\tau^{u,l}(n) - \hat{y}_\tau^{d,l}(n) < u_r$
                $\mathcal{D}_r \leftarrow D_r - x_\tau^s(n)$
            **End If**
        **End While**
**End For**
**Output:** Four trained functions $\{\mathbf{\Phi}_r, \mathbf{\Psi}_r, \mathbf{\Omega}_r, \mathbf{\Gamma}_r\}_{r=1:R}$ =0

---
**Algorithm 2** Deep Sensing - Testing Stage
---

**Input:** cost $C = \{c^1, c^2, ..., c^D\}$, corresponding trained functions $\mathbf{\Phi}_r, \mathbf{\Psi}_r, \mathbf{\Omega}_r, \mathbf{\Gamma}_r$, and $\lambda$
**For** $t \in \{1, 2, ..., T\}$
    **Interpolation:** $\tilde{x}_t^d = \mathbf{\Phi}\Big(\{x_\tau^d, m_\tau^d, \delta_\tau^d\}_{\tau=1:t-1}\Big)$

    **Imputation:** $\hat{x}_t^d = \mathbf{\Psi}\Big(\{x_t^d, \tilde{x}_t^d, m_t^d\}_{d=1:D}\Big)$

    **Error Estimation:** $\hat{e}_t^d = \mathbf{\Gamma}\Big(\{m_\tau^d\}_{d=1:D}, \{\delta_\tau^d, m_\tau^d\}_{\tau=1:t}\Big)$

        Compute the confidence interval of imputed value: $CI_x = (\hat{x}_t^d - \lambda\hat{e}_t^d, \hat{x}_t^d + \lambda\hat{e}_t^d)$
    **Prediction Interval:** Compute the confidence interval of predictions: $CI_y = (\hat{y}_t^{d,l}, \hat{y}_t^{d,u})$

        $\hat{y}^{d,l} = \min_{\hat{x}_\tau^d \in CI_x} \mathbf{\Omega}\Big(\{x_\tau^d, \hat{x}_\tau^d, \delta_\tau^d\}_{\tau=1:t,d=1:D}\Big)$
        $\hat{y}^{d,u} = \max_{\hat{x}_\tau^d \in CI_x} \mathbf{\Omega}\Big(\{x_\tau^d, \hat{x}_\tau^d, \delta_\tau^d\}_{\tau=1:t,d=1:D}\Big)$
    **Active Sensing:** Decision of sensing
        $a_t^d = \mathbb{I}(\hat{y}_t^{d,u} - \hat{y}_t^{d,l} > c^d)$
    **Prediction:** $\hat{y}_t = \mathbf{\Omega}\Big(\{x_\tau^d, \hat{x}_\tau^d, \delta_\tau^d\}_{\tau=1:t,d=1:D}\Big)$
**End For**
**Output:** Active sensing $\{a_t^d\}_{t=1:T,d=1:D}$, Prediction $\{\hat{y}_t\}_{t=1:T}$ =0

CONFIGURATIONS OF THE EXPERIMENTS

We used "interp1" package in MATLAB for interpolation algorithms, and "mice" and "amelia" packages in R for imputation algorithms. Table 5 illustrates the details of the prediction algorithm.

Table 5: Configurations of the Experiments

| Blocks | Categories | Configurations |
|---|---|---|
| **Interpolation** | Model
Initialization
Optimization
Batch size and iterations
Depth
Constraints | Modified Bi-RNN with GRU
Xavier Initialization (Glorot & Bengio (2010))
Adam* Optimization (Kingma & Ba (2014)) (learning rate = 0.05)
Batch size = 100, Iterations = 1000
5
The matrix parameters are block-diagonals |
| **Imputation** | Model
Initialization
Optimization
Batch size and iterations
Depth
Constraints | Fully Connected Layers
Xavier Initialization
Adam Optimization (learning rate = 0.05)
Batch size = 100, Iterations = 1000
10 (Activate function: Linear)
The block-diagonal part of the matrix is zero. |
| **Error Estimation** | Model
Initialization
Optimization
Batch size and iterations
Depth
Constraints | Multi-directional RNN (M-RNN) with GRU
Xavier Initialization
Adam Optimization (learning rate = 0.05)
Batch size = 100, Iterations = 1000
5 for Bi-RNN, 10 for FC with linear activation function
Block-diagonal matrix parameters for Bi-RNN |
| **Prediction** | Model
Initialization
Optimization
Batch size and iterations
Depth
Constraints | RNN with GRU
Xavier Initialization
Adam Optimization (learning rate = 0.05)
Batch size = 100, Iterations = 1000
10
None |

Adam*: Adaptive Moment Estimation

