# OpenReview forum: "Deep Sensing: Active Sensing using Multi-directional Recurrent Neural Networks"
_ICLR.cc/2018/Conference — Accept (Poster)_

### Official Review · AnonReviewer2 · 2017-11-27
**Interesting approach, but limited engagement with related work**

**Rating:** 6
**Confidence:** 3

**Review:**

This paper presents a new approach to determining what to measure and when to measure it, using a novel deep learning architecture. The problem addressed is important and timely and advances here may have an impact on many application areas outside medicine. The approach is evaluated on real-world medical datasets and has increased accuracy over the other methods compared against.

+ A key advantage of the approach is that it continually learns from the collected data, using new measurements to update the model, and that it runs efficiently even on large real-world datasets.

-However, the related work section is significantly underdeveloped, making it difficult to really compare the approach to the state of the art. The paper is ambitious and claims to address a variety of problems, but as a result each segment of related work seems to have been shortchanged. In particular, the section on missing data is missing a large amount of recent and related work. Normally, methods for handling missing data are categorized based on the missingness model (MAR/MCAR/MNAR). The paper seems to assume all data are missing at random, which is also a significant limitation of the methods.

-The paper is organized in a nonstandard way, with the methods split across two sections, separated by the related work. It would be easier to follow with a more common intro/related work/methods structure.

Questions:
-One of the key motivations for the approach is sensing in medicine. However, many tests come as a group (e.g. the chem-7 or other panels). In this case, even if the only desired measurement is glucose, others will be included as well. Is it possible to incorporate this? It may change the threshold for the decision, as a combination of measures can be obtained for the same cost.

---

> ### Author Response · Authors · 2017-12-14
> **Re: Interesting approach, but limited engagement with related work**
>
> Answer 1: As it is written, Deep Sensing applies if data is missing completely at random (MCAR) or just missing at random (MAR).  We will make this clearer in the revision.  The setting in which measurements are missing not at random (MNAR) is important but the literature dealing with this setting is small; see for instance the discussion in [1]. Deep Sensing can also be applied in the MNAR framework as well by incorporating the mask vector (which indicates missingness) as an additional input.  In the revised manuscript, we will discuss this point and provide additional experiments to highlight this point.
>
> [1] A. M. Alaa, S. Hu, and M. van der Schaar, "Learning from Clinical Judgments: Semi-Markov-Modulated Marked Hawkes Processes for Risk Prognosis," ICML, 2017
>
> Answer 2: We will revise the manuscript to conform to the more common format.
>
> Answer 3: Yes this can be incorporated easily.  For example: every set of tests that can be carried out as a single panel at the same cost can be considered as a single test.

---

### Official Review · AnonReviewer3 · 2017-11-29
**This paper proposes a novel method to solve the problem of active sensing**

**Rating:** 7
**Confidence:** 4

**Review:**

This paper proposes a novel method to solve the problem of active sensing from a new angle (Essentially, the active sensing is a kind of method that decides when (or where) to take new measurements and what measurements we should conduct at that time or (place)). By taking advantage of the characteristics of long-term memory and Bi-directionality of Bi-RNN and M-RNN, deep sensing can model multivariate time-series signals for predicting future labels and estimating the values of new measurements. The architecture of Deep Sensing basically consists of three components:
1. Interpolation and imputation for each of channels where missing points exist;
2. Prediction for the future labels in terms of the whole multivariate signals (The signal is a time-series data and made up of multiple channels, there is supposed to be a measured label for each moment of the signal);
3. Active sensing for the future moments of each of the channels.

Pros

The novelty of this paper lies in using a neural network structure to solve a traditional statistical problem which was usually done by a Bayesian approach or using the idea of the stochastic process.

A detailed description of the network architecture is provided and each of the configurations has been fully illustrated.  The explanation of the structure of the combined RNNs is rigorous but clear enough of understanding.

The method was tested on a large real dataset and got a really promising result based several rational assumptions (such as assuming some of the points are missing for evaluating the error of the interpolation & imputation).

Cons

How and why the architecture is designed in this way should be further discussed or explained. Some of the details of the design could be inferred indirectly. But somewhere like the structure of the interpolation in Fig.3 doesn't have any further discussion. For example, why using GRU based RNN, and how Bi-RNN benefits here.

---

> ### Author Response · Authors · 2017-12-14
> **Re: This paper proposes a novel method to solve the problem of active sensing**
>
> Answer 1: We will improve the explanation in two ways: (1) We will improve the discussion, going step-by-step. (2) We will follow a suggestion of Reviewer 1 and show how the accuracy of our method would be reduced if we carried out only imputation (no interpolation) or only interpolation (no imputation) or only operated in one direction (no bi-directionality).  We will also improve and clarify the discussion of active sensing.
>
> Answer 2: In many domains (e.g. the medical domain), the measurements display long-term correlations, and accurate prediction of current states requires capturing these long-term correlations. GRU-based RNN’s are well-known to be good for this purpose [1, 2]. Using a Bi-RNN rather than an ordinary unidirectional RNN (in the interpolation block) is important because it helps to capture the correlation of the current measurement with both previous and future measurements. To highlight these points, we will incorporate additional experiments in the revised manuscript that show the effects of Bi-RNN and GRU in comparison to the standard RNN framework.
>
> [1] Chung, J., Gulcehre, C., Cho, K., & Bengio, Y. Empirical evaluation of gated recurrent neural networks on sequence modeling. In NIPS 2014 Workshop on Deep Learning, 2014
>
> [2] Chung, J., Gulcehre, C., Cho, K., & Bengio, Y. Gated feedback recurrent neural networks. In International Conference on Machine Learning, 2015

---

### Official Review · AnonReviewer1 · 2017-11-30
**Intriguing angle on an under-studied problem in health+ML with strong empirical results**

**Rating:** 8
**Confidence:** 4

**Review:**

This is a very interesting submission that takes an interesting angle on clinical time series modeling, namely, actively choosing when to measure while simultaneously attempting to impute missing measurements and predict outcomes of interest. The proposed solution formulates everything as a giant learning problem that involves learning (a) an interpolation function that predicts a missing measurement from its past and present, (b) an imputation function that predicts a missing measurement from other variables at the same time step, (c) a prediction function that predicts outcomes of interest, including forecasting future measurements, (d) an error estimation function that estimates error of the forecasts in (c). These four pieces are then used in combination with a heuristic to decide when certain variables should be measured. This framework is used with a GRU-RNN architecture and in experiments with two datasets, outperforms a number of strong baselines.

I am inclined toward accepting this paper due to the significance of the problem, the ingenuity of their proposed approach, and the strength of the empirical results. However, I think that there is a lot of room for improvement in the current manuscript, which is difficult to read and fully grasp. This will lessen its impact in the long run, so I encourage the authors to strive to make it clearer. If they succeed in improving it during the review period, I will gladly raise my score.

NOTE: please do a thorough editorial pass for the next version -- I found at least one typo in the references (Yu, et al. "Active sensin.")

QUALITY

This is solid research, and I have few complaints about the work itself (most of my feedback will focus on clarity). I will list some strengths (+) and weaknesses (-) below and try to provide actionable feedback:

+ Very important problem that receives limited attention from the community
+ I like the formulation of active sensing as a prediction loss optimization problem
+ The learning problem is pretty intuitive and is well-suited to deep learning architectures since it yields a differentiable (albeit complex) loss function
+ The results speak for themselves -- for adverse event prediction in the MIMIC-III task, DS improves upon the nearest baseline by almost 9 points in AUC! More interestingly, using Deep Sensing to create a "resampled" version of the data set improves the performance of the baselines. It also achieves much more accurate imputation than standard approaches.

- The proposed approach is pretty complex, and it's unclear what is the relative contribution of each component. I think it is incumbent to do an ablation study where different components are removed to see how performance degrades, if at all. For example, how would the model perform with interpolation but not imputation? Is bidirectional interpolation necessary, or would forward interpolation work sufficiently well (the obvious disadvantage of the bidirectional approach is the need to rerun inference at each new time step). Is it necessary to use both the actual AND predicted measurements as inputs (what if we instead used actual measurements when available and predicted otherwise)?
- The experiments are thorough with a nice selection of baselines, but I wonder if perhaps Futoma, et al. [1] would be a stronger baseline than Choi, Che, or Lipton. They showed improvements over similar magnitude over baselines for predicting sepsis, and their approach (a differentiable GP-approximating layer) is conceptually simpler and has other benefits. I think it could be combined with the active sensing framework in this paper.
- The one question this framework appears incapable of answering in a straightforward manner is WHEN the next set of measurements should be made. One could imagine a heuristic in which predictive loss/gain are assessed at different points in the future, but the search space will be huge, particularly if one wants to optimize over measurements at different points, e.g., maybe the optimal strategy is to take roughly hourly vitals but no labs until 12 hours from now. Indeed, it might be impossible to train such a model properly since the sampling times in the available training data are highly biased.
- One thing potentially missing from this paper is a theoretical analysis to understand and analyze its behavior and performance. My very superficial analysis is that the prediction loss/gain framework is related to minimizing entropy and that the heuristic for choosing which variables to measure is a greedy search. A theoretical treatment to understand whether and how this approach might be sub-optimal would be very desirable.
- Are the measurement and prediction "confidence intervals" proper confidence intervals (in the formal statistical sense)? I don't think so -- I wonder if there are alternatives for measuring uncertainty (formal CIs or maybe a Bayesian approach?).

CLARITY

My main complaint about this paper is clarity -- it is not difficult to read per se, but it is difficult to fully grok the details of the approach and the experimental setup. From the current manuscript, I do not feel confident that I could re-implement Deep Sensing or reproduce the experiments. This is especially important in healthcare research, where there is a minor reproducibility crisis, even for resarch using MIMIC (see [2]). Of course, this can be alleviated by publishing the code and using a public benchmark [3], but it can't hurt to clarify these details in the paper itself (and to add an appendix if length is an issue).

Here are some potential areas for improvement:

- The structure of the paper is a bit weird. In particular section 2 (pages 2-4) seems to be a grab bag of miscellaneous topics, at least by the headers. I think the content is fine -- perhaps section 2 can be renamed as "Background," subsection 2.1 renamed as "Notation," and subsection 2.2 renamed as "Problem Formulation" (or similar). I'd just combine subsection 2.3 with the previous one and explain that Figure 1 illustrates the problem formulation.
- The active sensing procedure (subsection 2.2, page 3, equation 1 and the equations just above) is unclear. How are the minimization and maximization performed (gradient descent, line search, etc.)? How is the search for the subset of measurement variables performed (greedy search)? The latter is a discrete search, and I doubt it's, e.g., submodular, so it must be a nontrivial optimization.
- Related, I'm a little confused about equation 1: C_T is the set of variables that should be measured, but C_T is being used to index prediction targets -- is this a typo?
- The related work section is pretty extensive, but I wonder if it should also include work on active learning (Bayesian active learning, in particular, has been applied to sensing), submodular optimization (for sensor placement, which can be thought of as a spatial version of active sensing), and reinforcement learning.
- I don't understand how the training data for the interpolation and imputation functions are constructed. I *think* that is what is described in the Adaptive Sampling subsection on page 8, but that is unclear. The word "representations" is used here, but that's an overloaded term in machine learning, and its meaning here is unclear from context. It appears that maybe there's an iterative procedure in which we alternate between training a model and then resampling the data using the model -- starting with the full data set.
- The distinction between training and inference is not clear to me, at least with respect to the active sensing component. Is selective sampling performed during training? If so, what happens if the model elects to sample a variable at time t that is not actually measured in the data?
- I don't follow subsection 4.2 (pages 8-9) at all -- what is it describing? If by "runtime" the authors refer to the computational complexity of the algorithm, then I would expect a Big-O analysis (none is provided -- it's just a rather vague discussion of what happens). I'd recommend removing this entire subsection and replacing it with, e.g., an Algorithm figure with pseudocode, as a more succinct description.
- For the experiments, the authors provide insufficient detail about the data and task setup. Since MIMIC is publicly available, then readers ought (hypothetically) to be able to reproduce the experiments, but that is not currently possible. As an example, what adverse events are being predicted? How are they defined?
- Figure 4 is nice, but it's not immediately obvious what the connection between observation rate and sampling cost. The authors should explain how a given observation rate is encoded as cost in the loss function.

ORIGINALITY

While active sensing is not a new research topic per se, there has been very limited research into the specific question of choosing what clinical variables to measure when in the context of a given prediction problem. This is a topic that (in my experience) is frequently discussed but rarely studied in clinical informatics circles. Hence, this is a very original line of inquiry, and the prediction loss/gain framing is a unique angle.

SIGNIFICANCE

I anticipate this paper will generate significant interest and follow-up work, at least among clinical informaticists and machine learning + health researchers. The main blockers to a significant impact are the clarity of writing issues listed above -- and if the authors fail to publish their code.

REFERENCES

[1] Futoma, et al. An Improved Multi-Output Gaussian Process RNN with Real-Time Validation for Early Sepsis Detection. MLHC 2017.
[2] Johnson, et al. Reproducibility in critical care: a mortality prediction case study. MLHC 2017
[3] Harutyunyan, et al. Multitask Learning and Benchmarking with Clinical Time Series Data. arXiv.

---

> ### Author Response · Authors · 2017-12-14
> **Re: Intriguing angle on an under-studied problem in health+ML with strong empirical results**
>
> Answer 1: We will carefully revise the entire paper.
>
> Answer 2: These are useful suggestions and we will follow them. To highlight the source of gain, we will carry out several additional experiments and report the results in the revised manuscript. In particular, we will carry out an experiment in which the model is restricted to interpolation (no imputation), another experiment in which the model is restricted to imputation (no interpolation) and a third experiment in which only forward interpolation (no backward interpolation) is performed.
>
> Answer 3: Indeed this is exactly what we are doing: we use actual measurements when available and predicted measurements when actual measurements are not available.  We will make this clearer in the revision.
>
> Answer 4: This is also a good suggestion.  In the revision, we will add discussion of Futoma, et al in the related works and will conduct experiments to compare the performance with that of Deep Sensing in various settings.
>
> Answer 5: Deep Sensing does answer the question of when the next set of measurements should be made.  At each time T, Deep Sensing asks whether there are any measurements to be made at time T+1 for which the benefit outweighs the cost.  If the answer is “yes” then Deep Sensing recommends that those measurements should be made at time T+1.  If the answer is “no” then Deep Sensing asks whether there are any measurements to be made at time T+2 for which the benefit outweighs the cost, and so forth. Thus Deep Sensing is recommending both a time at which the next measurements should be taken and which measurements should be taken at that time.
>
> This is a greedy procedure, and it is conceivable that although it is beneficial to take measurements at time T+1, it would be even more beneficial to wait and take measurements at time T+2 instead.  Deep Sensing could be modified to be forward-looking (rather than greedy), but this would expand the search space enormously.  We will add a discussion of this point.
>
> Alternatively, one can imagine asking Deep Sensing to decide at time T whether to take one set of measurements at time T+1 and a second set of measurements at time T+2 and a third set of measurements at time T+3, and so forth. However, it is not clear what advantage would be obtained by doing this.  As currently formulated, Deep Sensing can decide at time T to take a set of measurements at time T+1, and then at time T+1 – after the results of those measurements become available – it can decide what measurements to take at time T+2, and so forth. We will add a discussion of this point as well.
>
> Answer 6: We will add a more theoretical treatment as suggested.

---

> ### Author Response · Authors · 2017-12-14
> **Re2: Intriguing angle on an under-studied problem in health+ML with strong empirical results**
>
> Answer 7: We think that modulo some (reasonable) assumptions, we do in fact use a proper confidence interval. However, we agree that more justification/discussion
> is warranted, and we will add it in the revised manuscript, along the line sketched below.
>
> Define \hat{x} = x + n. For the moment, assume that n is Gaussian noise and
> that we can interpret the error e = | \hat{x} - x | as the estimated standard deviation of the Gaussian noise n. Then, (\hat{x} - \lambda \times e, \hat{x} + \lambda \times e) is the proper confidence interval for x in the formal statistical sense.
>
> The assumption of Gaussian noise is quite standard and probably needs no further comment. The interpretation of the error as the estimated standard
> deviation of Gaussian noise is not standard but can be justified in the following
> way. If our estimate \hat{x} is the expected value of x, then we will have x = \hat{x} + n, where x is the observed measurement from a Gaussian distribution, \hat{x} is the expected value of x and n is normal (Gaussian). In that case, the expected value of e = | \hat{x} – x | = | n | is just the standard deviation of Gaussian noise, which is \sqrt{E[n^2]}. Hence, we need two assumptions: (1) our estimate \hat{x} is the expected value of x; (2) the observed measurement can be approximated as the sum of the expectation of x and Gaussian noise (approximate normality [1, 2, 3, 4]).
>
> To justify these assumptions, we proceed as follows. Assume that the measurement x is sampled from an unknown distribution P_\theta; i.e. x ~ P_\theta. If P_\theta is itself normal (Gaussian), then it follows that.
>
> x ~ P_\theta  \Leftrightarrow  x = E[x] + n
>
> where n ~ N(0, \sigma^2). (This uses the observation that the expectation of the normal distribution is the mean). In general, we cannot assume that P_\theta is normal, but it will be enough if it is approximately normal, which is a common assumption in the literature (see [1, 2, 3, 4] for instance.)  In that case, following the literature we can obtain
>
> x \approximate E[x] + n
>
> where n ~ N(0, E[(x-E[x])^2]). From this we obtain that \hat{x} = E[x].  (In practice, we use \hat{x} as the sample mean of x which converges to E[x]). In that case the distribution of the error e = |\hat{x} – x| coincides with the distribution of the absolute value of samples generated by the normally distributed noise n:
>
> E[e] = E[|\hat{x}-x|] = E[\sqrt{(\hat{x}-x)^2}]=E[\sqrt{n^2}] = E[\sqrt{(x-E[x])^2}]).
>
> Thus, estimating e can be interpreted as estimating the standard deviation of the noise.
>
> [1] Rothenberg, Thomas J. "Approximate normality of generalized least
> squares estimates." Econometrica: Journal of the Econometric Society (1984):
> 811-825.
> [2] Davison, A. C., and D. V. Hinkley. "Bootstrap Methods and Their Application, Cambridge Univ." Press, Cambridge (1997).
> [3] Efron, Bradley, and Robert Tibshirani. "Bootstrap methods for standard errors, confidence intervals, and other measures of statistical accuracy." Statistical science (1986): 54-75.
> [4] Bartlett, Maurice S. "Approximate confidence intervals. II. More than one unknown parameter." Biometrika 40.3/4 (1953): 306-317.
>
> Answer 8: We will of course publish the code in the Github after the paper is accepted.  We will make every effort to clarify both the approach and the experimental aspects.
>
> Answer 9: These are good suggestions and we will follow them.
>
> Answer 10: With respect to the equations just above equation (1): We carry out the minimization and maximization by one-dimensional gradient descent.  This is possible because the minimization and maximization problems for each feature are independent.  With respect to the maximization in equation (1): What we wrote was (accidentally) misleading and we will correct it in the revision.  If the number of possible measurements is large, and there are complementarities among measurements, then the actual optimization problem requires examining all possible subsets of measurements – which is an intractable problem.  Instead, we follow a greedy procedure: we identify all the measurements with the property that the value of that measurement (by itself) exceeds its cost, and we take C^*_{T+1} to be that set of measurements.  Thus we solve a tractable optimization problem that yields an approximation to the actual optimal set of measurements. We will clarify this in the revision.

---

> ### Author Response · Authors · 2017-12-14
> **Re3: Intriguing angle on an under-studied problem in health+ML with strong empirical results**
>
> Answer 11: We will check carefully but that was certainly not intended.  C_T is used to index prediction targets; the argmax is C^*_T.
>
> Answer 12: We will add the suggested related works (active learning, submodular optimization, and reinforcement learning) in the revised manuscript.
>
> Answer 13: We agree that the meaning of “representations” in this context is unclear and we will not use it in the revision.  The training procedure is as follows. In the first step, we train the M-RNN architecture (the interpolation and imputation functions) using the original data set. In the second step, we fix a threshold and delete measurements (resampling the original data set) whose estimated “information gain – cost” is smaller than the fixed threshold; this procedure yields a resampled data set. In the third step, we re-train the M-RNN architecture using the resampled data set. We then increase the threshold and repeat the second and third steps, continuing through whatever set of thresholds are chosen.  We will clarify this in the revision.
>
> Answer 14: If the actual dataset is complete this is of course not a problem.  If the actual dataset is not complete, we only consider measurements that are actually recorded in the dataset. For example, suppose the dataset records vital signs every hour but lab tests only every 12 hours and we are at a time T when both vital signs and lab tests are recorded.  In determining what – if anything should be sampled one hour later, we consider only vital signs and not lab tests. Of course, after 11 more hours have elapsed, then lab tests at the next hour are possible and considered.  We will clarify this in the revision.
>
> Answer 15: We agree.  We will remove this subsection and replace it with the pseudocodes for the algorithms.
>
> Answer 16: We will clarify the discussion of the experiment; in particular we will clarify the discussion of the adverse events that are being predicted in each case (mortality in MIMIC-III dataset and admission to the ICU in the Wards dataset).
>
> Answer 17: The cost of each possible measurement is well-defined. If all measurements were equally costly, we could identify the cost with the observation rate.  If some measurements are most costly than others, we weight those measurements more heavily when expressing the cost in terms of the observation rate.  We will explain this more thoroughly in the revision.
>
> Answer 18: Of course, we will publish our code in Github. However, because the review process is anonymous, we will publish our code after the final decision.

---

> ### Comment · AnonReviewer1 · 2018-01-14
> **Strong response and revision, raised score to 8**
>
> I have raised my score from 6 to 8 to reflect the authors' thorough responses and significant improvements in the revised manuscript.
>
> From my own review, the authors addressed the following points in their revision:
>
> - expanded related work to cover active learning and submodular optimization
> - added a discussion of AND empirical comparison with Futoma [1] -- though I should note that I find the poor performance of Futoma on the MIMIC-III data set surprising (I wonder about hyperparameter tuning for your Futoma implementation)
> - improved clarify of the paper, including an explicit discussion of how the algorithm can be used to decide WHEN to measure and the fact that it uses a *greedy* search strategy, the details of the prediction tasks in the experiments, and the way in which the subsampled data set was created
> - added an ablation study to understand the relative contribution of each model component (imputation, interpolation, backward imputation, GRU hidden layer, etc.)
> - added algorithm pseudocode and discussion of the approximate confidence intervals to the appendix
>
> This is cool work -- I look forward to the eventual code release to try it out.

---

### Decision · Program_Chairs · 2018-01-29
**ICLR 2018 Conference Acceptance Decision**

**Decision:**

Accept (Poster)

**Comment:**

This paper is well written, addresses and interesting problem, and provides an interesting solution.